# Comparative study on corrosion characteristics of conductive concrete in red soil environment

**Shan Lin[1], Xu Tian[2], Feng Pei[2], Pinghao Yang[3], Hongbo Cheng[3]***

**1** Guangzhou Metro Design&Research Institute Co.,Ltd., Guangzhou, Guangdong province, China, **2** State Grid Jiangxi Electric Power Co., Ltd. Electric Power Research Institute, Nanchang, Jiangxi province, China, **3** School of Electrical and Automation Engineering, East China Jiaotong University, Nanchang, Jiangxi province, China

* waitingbo@126.com

**Data Availability Statement:** All excel files are available from the figureshare database (accession number(s) https://doi.org/10.6084/m9.figshare.26968486.v1).

## Abstract

In order to solve the corrosion problem of grounding materials in highly corrosive red soil environments, conductive concrete was proposed as a new type of grounding material. The corrosion resistance of conductive concrete was tested and compared to select a suitable preparation scheme with excellent corrosion resistance. A series of conductive concrete samples were made using different conductive materials such as graphite, stainless steel fiber (SSF), and ordinary silicate concrete. Common grounding metals include Q235 steel and galvanized steel, embedded in red soil, conventional concrete, and conductive concrete. The open circuit potential, dynamic potential polarization, and electrochemical impedance spectroscopy of these metals were measured and analyzed. The open-circuit potential of metal electrodes in red soil is lower than that in concrete, and the potential of specimens with conductive phase is higher than that of the ordinary concrete, show that the corrosion sensitivity of metals in conductive concrete is greatly reduced, the corrosion potential is the lowest and the corrosion current is the highest in red soil, the capacitance arc radius in red soil is very small, indicating poor corrosion resistance of grounding metals in a red soil environment. All these indexes indicate that using conductive concrete as grounding material can effectively slow down the corrosion of grounding materials in red soil. Compared with stainless steel based conductive concrete, graphite based conductive concrete provides better protection for the grounding metal wrapped around it. As the content of conductive phase materials increases, the corrosion tendency and corrosion rate of conductive concrete decrease. Compared with Q235 carbon steel, galvanized steel exhibits excellent corrosion resistance in conductive concrete. Therefore, it is recommended to use galvanized steel conductive concrete as the grounding material for power systems in red soil environments.

## 1. Introduction

Transmission tower grounding is responsible for the safe operation of power system. Since the grounding metal work in an outdoor environment for a long time, it is prone to corrosion due

**Funding:** This research was funded by the Key projects of Natural Science Foundation of Jiangxi Province under grant 2021ACB204004, the funder provided funding for this study, and with the funding, we conducted relevant experiments. The funder Hongbo Cheng designed the experiment, analyzed the data and prepared the manuscript.

**Competing interests:** The authors have declared that no competing interests exist.

to the influence of environment such as soil, air and rainwater [1, 2]. The deterioration of the corrosion degree will lead to surface corrosion for metal transmission towers, which will reduce the mechanical strength of the towers and may endanger the safety of the whole transmission line. At the same time, it will also lead to the fracture of the grounding metal and the increase of the grounding resistance [3]. In many areas with poor soil conditions, grounding grids often fail to reach a safe service lifetime. The corrosion of grounding metal has become one of the important problems affecting the safe operation of power system [4, 5].

Red soil is widely distributed on Earth, it is a kind of soil formed under subtropical bioclimatic conditions. There are significant differences in soil characteristics between red soil and other types of soil, the comparison of parameter indicators between red soil and other types of soil is shown in Table 1 [6].

Table 1 shows that the pH value, moisture content, resistivity, and salt content of red soil differ from those of other soil types. Red soil has a low pH value, indicating it is acidic. This acidity increases the H+ concentration in the soil water, which dissolves the protective film on the surface of metals, leading to a significant increase in the metal corrosion rate. Red soil has a high moisture content, which can be up to 30%, much higher than other types of soil. The abundant water in red soil can create good conditions for electrochemical corrosion of buried metals, which will accelerate the corrosion rate of metals in red soil. The resistivity of red soil is low, the ion exchange is less hindered in red soil, resulting in a higher current in the corrosive circuit and a stronger corrosion effect on metals.

In summary, the low pH value, high water content, and low electrical resistivity of red soil are all beneficial for the development of buried metal corrosion, therefore, the corrosion of grounding metals is particularly severe in red soil environments. To avoid grounding failure caused by corrosion, it is necessary to search for new grounding materials to ensure good and normal operation in highly corrosive red soil environments [7].

Conductive concrete has both the bonding properties of ordinary concrete and the conductivity of conductors, it can be used as a grounding material [8], Conductive concrete has been used as a new grounding material and its corrosion characteristics have been tested in the special environment of red soil, the effects of conductive phase materials and content on the corrosion resistance of conductive concrete are analyzed, and conductive concrete ratio schemes with good corrosion resistance in red soil environments are screened out, thus providing a new material for grounding in red soil environments.

## 2. Related works

The common resistance reduction methods of grounding grid include extending the area of grounding grid, increasing the depth of grounding grid and improving soil quality, but the corrosion of grounding metal is always inevitable [9, 10]. Extending the area of grounding grid and increasing the depth of grounding grid increase the risk of corrosion of grounding metals, because they both increase the amount of grounding metals, and there may be metal in a locally strong corrosion environment. The use of resistance reducing agent to improve soil

**Table 1. Characteristic parameters of red soil and several typical soils.**

| soil | pH value | moisture content /% | resistivity/Ω·m | salt content /% |
|---|---|---|---|---|
| Red soil | 4.5–5.5 | 15–25 | 628–1200 | 0.0059–0.0129 |
| desert soil | >8.0 | 0.1–1.7 | 24–49 | 0.055–0.721 |
| saline-alkali soil | 7.5–10 | 0.23–0.8 | 0.1–0.7 | 0.3–1.9669 |
| alpine soil | 6.5–8.0 | 3.2–5.4 | 29–40 | 0.041–0.052 |

quality will accelerate the corrosion rate of grounding metal because of its strong corrosion effect. For example, after 1.4 years of burial in acidic soil, the corrosion rate of Q235 carbon steel reaches 6-8g/dm$^2$·a, the pitting corrosion rate is 0.7–1.3mm/a (corrosion grade is classified as grade 5). The corrosion pits on the surface of the samples are dense and continuous, and the corrosion is serious [11].

The development of electric power systems has seen significant advancements in recent years, with a key focus on improving safety, reliability, and efficiency. One pivotal area of innovation is the introduction of new materials for grounding systems, which play a critical role in electrical infrastructure. For instance, [12] conducted a comprehensive study on the corrosion resistance of non-metallic grounding materials, highlighting their durability in harsh environmental conditions. Additionally, [13] explored the mechanical properties of conductive polymers and their potential applications in grounding systems, shedding light on their structural advantages. These findings collectively underscore the significance of this new trend in electric power systems.

These non-metallic grounding materials generally use graphite, carbon black, metal fibers, carbon fibers, polyaniline and other materials to prepare resin-based conductive coatings or conductive colloids: conductive anticorrosive coatings are generally coated on the surface of metal grounding bodies to isolate the electrochemical corrosion path of metals and soil electrolytes, and similar to electroplated alloy materials [14–16]. To prevent the expansion of oxygen concentration corrosion, there are strict requirements for uniformity of conductive coating. This method is suitable for the coating before the new grounding grid enters the ground. For the reconstruction of the grounding grid in operation, the practical construction of this method is more difficult, and for the long-term effectiveness and anti-aging ability of the conductive anti-corrosion adhesive, there is still a lack of convincing practical operation experience data. From the current use of non-metallic grounding materials, it is generally used as auxiliary peripheral grounding materials of metal grounding grids, combined with metal grounding materials, to achieve the purpose of resistance reduction and corrosion prevention [17]. Considering the actual grounding engineering of transmission line tower grounding grid, steel and galvanized steel (copper clad steel under extreme geological conditions) are still the main consideration for material cost.

In recent years, the application of concrete in power industry mostly stays in the application level of engineering buildings, only using the characteristics of good mechanical strength of concrete, the electrical performance of concrete has not been improved too much [18, 19]. Some scholars have developed a flexible graphite composite grounding material, which can avoid metal corrosion by replacing the grounding material. Its resistivity can be as low as 3.25*10$^{-5}$Ω·m [20, 21]. However, due to the limitation of graphite material properties, its mechanical properties are worse than galvanized steel and other electroplated alloy materials.

Conductive concrete is a remarkable composite material that merges the structural advantages of conventional cement with the unique property of electrical conductivity. This innovative blend of materials has garnered significant attention in recent years due to its wide-ranging applications and potential to revolutionize various industries [22]. According to the characteristics of conductive, electrothermal and electromagnetic shielding of conductive concrete, it has been used in ice melting, structural health detection and electromagnetic shielding [23–25], it is suitable for the grounding of power towers too, because it has good conductivity and high mechanical strength. [26] made a investigation of the conductive concrete to replace the conventional grounding system of substation, the grid resistance, ground potential, step and touch voltage have been simulated, the numerical results indicated that conductive concrete can perform better than conventional metal rods. In [27], a computational model of grounding resistance for conductive concrete stereo grounding was established, and a laying

scheme for electrically conductive concrete rod grounding electrodes in different soil types was designed. These studies demonstrate the significant application potential of conductive concrete in power system grounding. However, the anti-corrosion performance of conductive concrete grounding remains under-researched.

## 3. Grounding metal corrosion evaluation index and test method

### 3.1. Corrosion principle of grounding metal in red soil

Grounding metal in the soil is influenced by complex soil factors such as temperature, humidity, and pH value, making surface corrosion inevitable. Common types of corrosion include microbattery corrosion and electrolytic corrosion.

Generally, soil moisture contains a large number of electrolytes, such as acids, alkalis, and salts, which can cause a microbattery reaction with the buried grounding metal, leading to the corrosion of the metal surface [15]. For example, for the commonly used grounding material galvanized steel, the zinc on its surface will lose electrons in the electrolyte and form soluble zinc ions. The reaction is as follows:

$$Zn - 2e^- \rightarrow Zn^{2+} \tag{1}$$

When electrons enter the solution in an acidic environment, they react with hydrogen ions to generate hydrogen. In a neutral or alkaline environment, the electrons combine with oxygen in the solution to form hydroxide ions, resulting in oxygen absorption corrosion:

$$Acid: \quad 2H^+ \xrightarrow{+2e^-} H_2 \uparrow \tag{2}$$

$$Alkaline: \quad O_2 + 2H_2O \xrightarrow{+4e^-} 4OH^- \tag{3}$$

In this process, the movement of electrons generates a current between impurities and zinc metal, creating a battery effect. The essence of the micro battery corrosion mechanism is that two different micro components are in direct contact, in which the electrolyte is used as a medium to dissolve part of the anode metal.

When the grounding metal of the power system releases lightning strike current, short-circuit current or stray current, the electrons generated by these currents will accelerate electrochemical corrosion and turn the micro cell reaction that has occurred into electrolytic cell corrosion. In this process, the anode has a low potential and the cathode has a high potential. The reaction principle and products are similar to those of micro cell corrosion.

### 3.2. Grounding metal corrosion evaluation index

Open circuit potential, potentiodynamic polarization curve and electrochemical impedance spectroscopy are commonly used to evaluate metal corrosion, they can comprehensively reflect the corrosion characteristics of metals.

**(1) Open-circuit potential.** Open circuit potential (OCP) refers to the electrode potential measured when the current density is zero; it is the potential difference between the working electrode and the reference electrode when there is no load. A larger potential difference indicates a greater corrosion tendency. Thus, OCP can describe the process of an electrode transitioning from instability to stability. Its numerical value reflects the corrosion tendency of a metal material under different environmental conditions [15, 17–19].

**(2) Potentiodynamic polarization curve.** Potentiodynamic polarization method is a voltage current relationship curve obtained by scanning from the starting potential to the ending

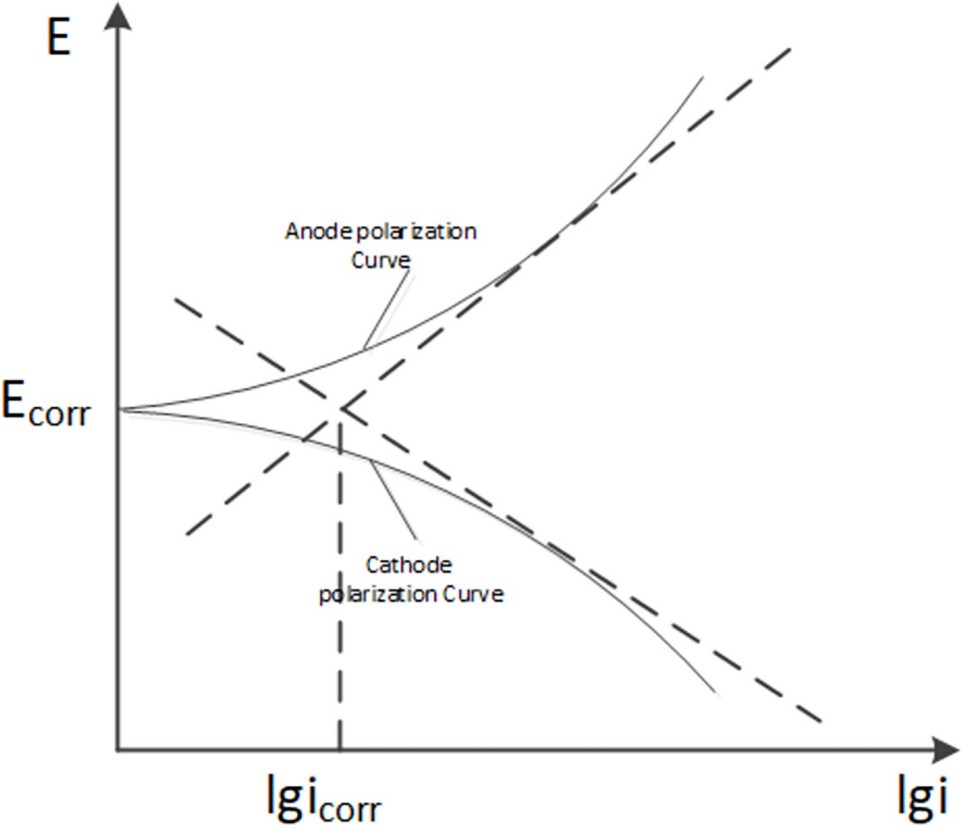

**Fig 1. Tafel curve of potentiodynamic polarization.**

potential according to the given step time and step height. The corrosion rate can be calculated by extrapolating the Tafel linear segment of the potentiodynamic polarization curve to the natural corrosion potential [28].

Tafel equation is used to describe the relationship between electrode potential and applied polarization current density:

$$|\Delta E| = -b_A \lg i_{corr} + b_A \lg i_{A_E} \tag{4}$$

$$|\Delta E| = -b_c \lg i_{corr} + b_c \lg i_{c_E} \tag{5}$$

Where, $\Delta E$ is the potential difference between the mental and the reference electrode, $b_A$ is the slope of the anode polarization curve in the linear relationship, $b_C$ is the slope of the cathode polarization curve in the linear relationship, and $i_{corr}$ is the current of the intersection point of the reverse extension lines of the two tangents.

Fig 1 displays both the polarization curve and the cathode polarization curve. In the semilogarithmic coordinate system, the polarization curve exhibits linearity in the strong polarization region, and its intersection with the horizontal axis allows for the calculation of the corrosion current.

**(3) Electrochemical Impedance Spectroscopy (EIS).** Electrochemical impedance spectroscopy measures the ratio of AC signal voltage to current by applying a small amplitude AC signal across different frequencies to the electrochemical system. This technique is valuable for describing the corrosion characteristics of metals [22, 29].

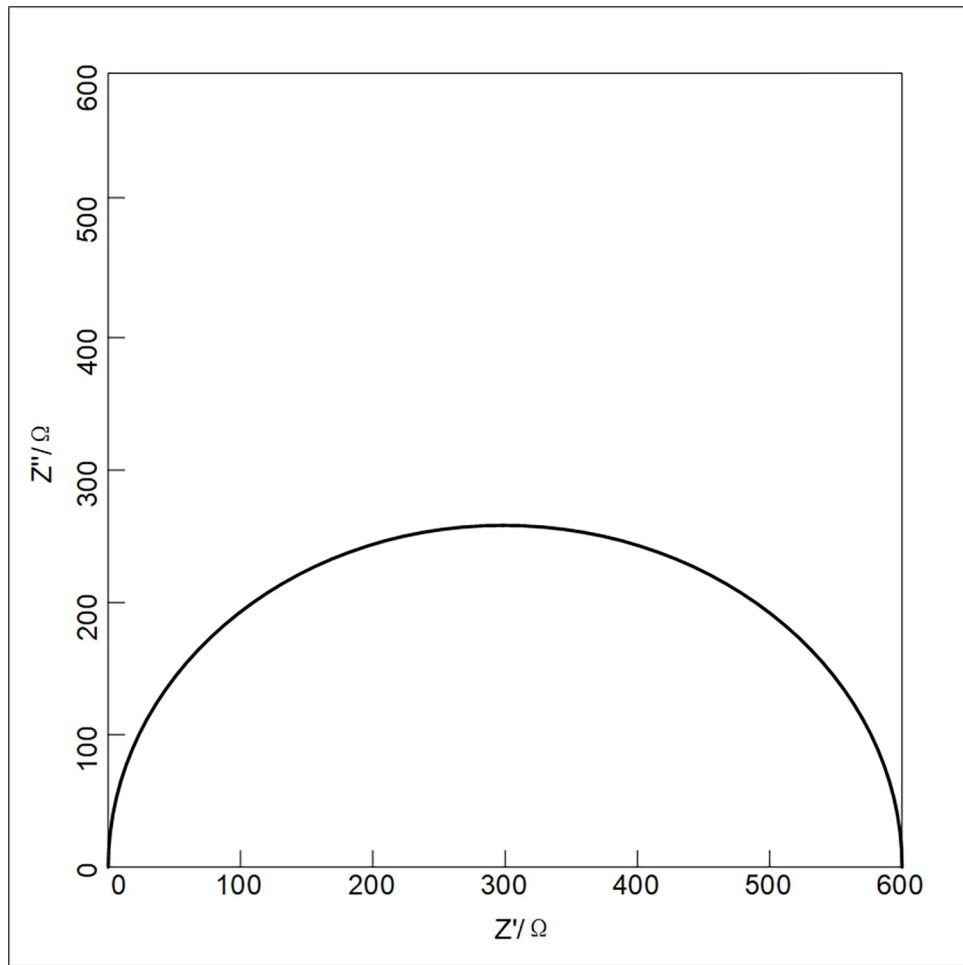

**Fig 2. Typical electrochemical impedance spectroscopy of metal surface without any passivation layer.**

Fig 2 is a typical electrochemical impedance spectroscopy of metal surface without any passivation layer, and Fig 3 is a typical electrochemical impedance spectroscopy of metal surface with a dense passivation layer. It can be seen from these two Figs that for the metal with severe corrosion, the capacitive arc radius of EIS is small, while for the metal coated by the dense passivation film, EIS has a large capacitive arc.

Therefore, the impedance spectrum can be used to measure the polarization resistance. According to the Nyquist curve, if the capacitance arc radius is larger, the polarization resistance is larger, and the corrosion resistance of the metal is better [24, 25]. In addition, the process of corrosion can also be judged by the shape of electrochemical impedance spectroscopy.

### 3.3. Grounding metal corrosion test scheme

The electrochemical workstation has the functions of potentiostat and galvanostat, which is very convenient for the measurement and analysis of electrochemical test data. The scheme for conducting electrochemical corrosion test on conductive concrete samples using electrochemical workstation is shown in Fig 4.

In Fig 4, the conductive concrete sample corresponds to the working electrode in the three electrode system and can be placed together with the grounding metal material to be tested.

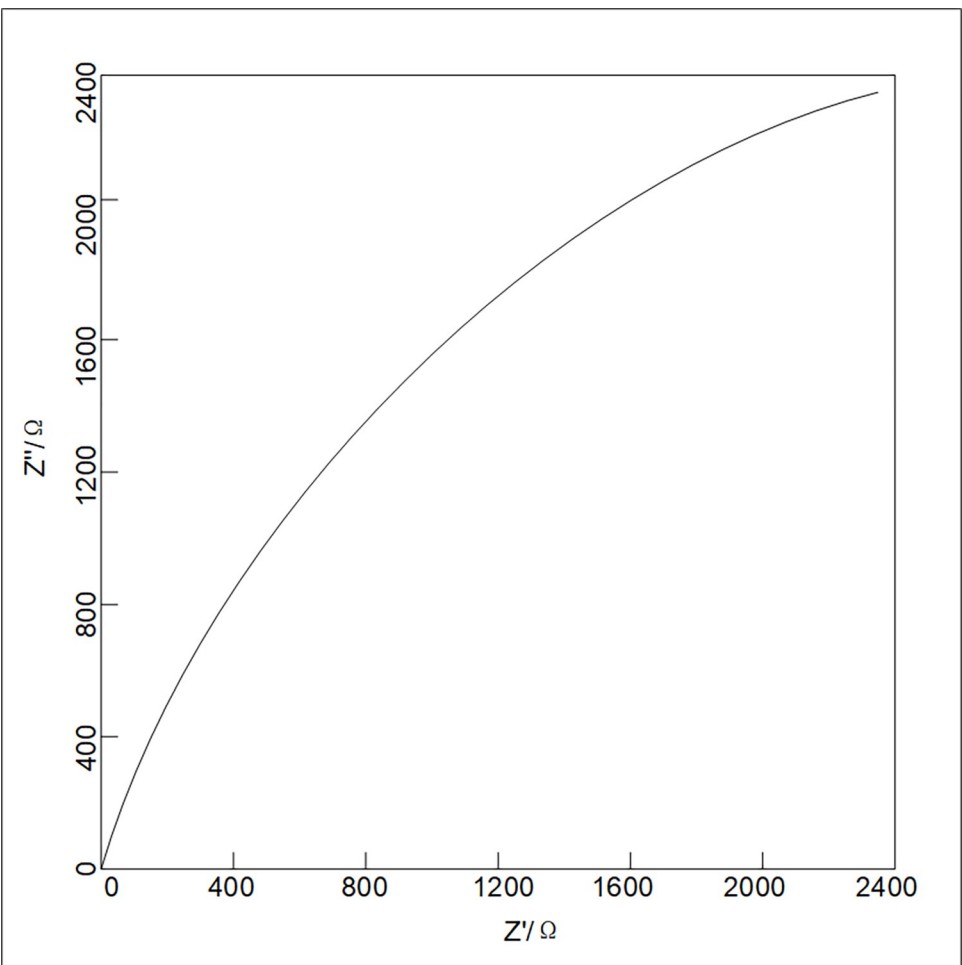

**Fig 3. Typical electrochemical impedance spectroscopy of metal surface with a dense passivation layer.**

The reference electrode is calomel electrode; The auxiliary electrode is The reference electrode used is a calomel electrode, while the auxiliary electrode is a platinum electrode. The concrete used is conductive concrete. The copper wire is extracted from the metal sample, and the exposed portion is encapsulated with epoxy resin.

Due to the slow variation of open-circuit potential, the recording frequency is set to 5 Hz, corresponding to every 0.2 seconds. The potentiodynamic scan rate is 0.5 mV/s, spanning from -0.2 V to the positive direction. The sinusoidal AC signal amplitude applied is 10 mV, with frequencies measured ranging from 0.01 Hz to 100 kHz during the entire impedance spectrum test. These tests are conducted under room temperature in an open environment.

## 4. Testing and analysis of corrosion characteristics of conductive concrete grounding materials

To compare the corrosion characteristics of different metals in different environments, the experimental plan is shown in Fig 5. Q235 carbon steel and stainless steel were used to test their corrosion characteristics in ordinary concrete, conductive concrete, and red soil environments, respectively. Specifically, there are two types of conductive concrete: one containing 30%, 35%, 40%, 50%, and 55% graphite, and the other containing 1.0%, 1.5%, 2.0%, 2.5%, and 3.0% stainless steel fibers.

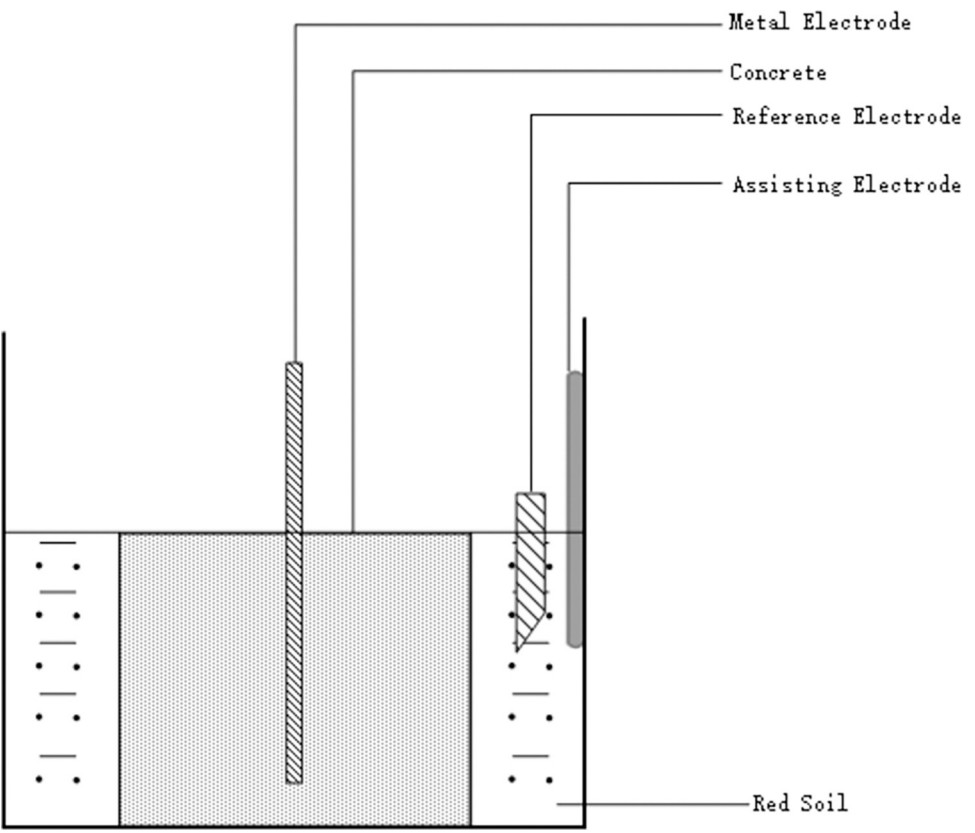

**Fig 4. Experimental schematic diagram.**

## 4.1. Preparation of conductive concrete specimens

In order to improve corrosion resistance while ensuring conductivity, concrete is used as the framework material, graphite and stainless steel fibers are added as conductive materials to prepare conductive concrete, the conductive material is fully mixed with cement and fine sand, after natural solidification, demoulding and maintenance for 28 days, the conductive concrete is finally made. Graphite is chosen as the representative of non-metallic conductive materials, and stainless steel fiber is chosen as the representative of metallic conductive materials, 3 sets of graphite conductive concrete samples with a mass content of 30%,35%,40%,50% and 55% were prepared, as well as 3 sets of stainless steel fiber conductive concrete samples with a mass content of 1.0%、1.5%、2.0%、2.5%、3.0% respectively.

The prepared conductive concrete sample is shown in Fig 6, and the concrete sample with the tested metal pre inserted in the test is shown in Fig 7.

The CH310H electrochemical workstation is used to form the test platform shown in Fig 8 through connection according to the test schematic diagram shown in Fig 4.

## 4.2. Grounding metals corrosion characteristics testing in different environments

In this study, we investigated the corrosion characteristics of two common grounding metals, Q235 carbon steel (comprising C, Mn, Si, S, P, and Fe) and galvanized steel (composed of common carbon steel with a zinc and aluminum coating), in various environmental conditions, including ordinary concrete, conductive concrete, and red soil environments. The conductive

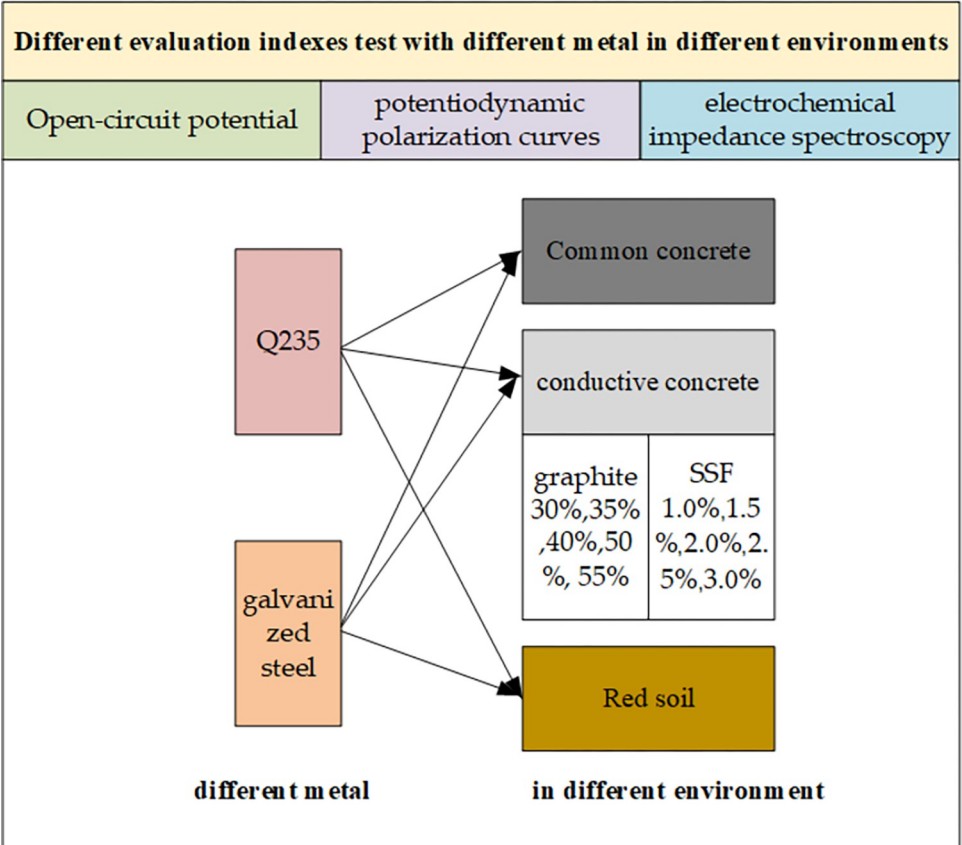

**Fig 5. Diagram of experimental plan.**

concrete was further categorized into two types: graphite conductive concrete with graphite content levels of 30%, 35%, 40%, 50%, and 55%, and stainless steel fiber conductive concrete with stainless steel fiber content levels of 1.0%, 1.5%, 2.0%, 2.5%, and 3.0%. An electrochemical

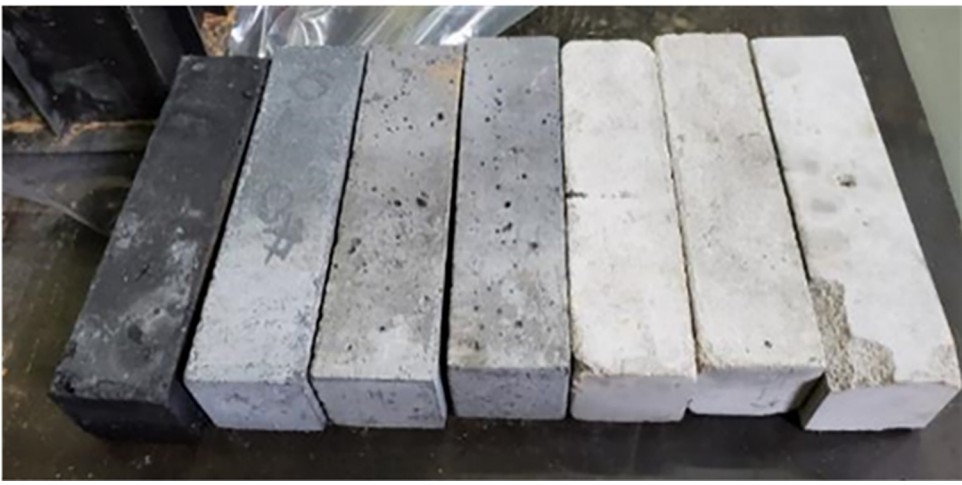

**Fig 6. Concrete specimens with different conductive materials.**

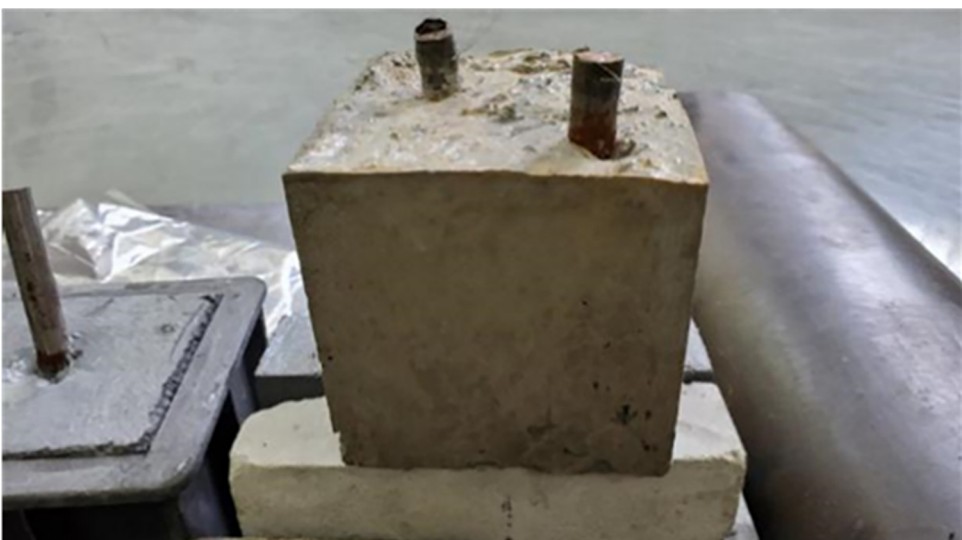

**Fig 7. Conductive concrete specimens pre-positioned with metal.**

workstation (CS310H) was employed to measure the open circuit potential (OCP), polarization curves, and electrochemical impedance spectroscopy (EIS) of both metals under these conditions. The aim was to comprehensively compare and analyze the corrosion behavior of these metals across different environmental settings.

The results from the electrochemical tests revealed distinct corrosion characteristics for Q235 carbon steel and galvanized steel in the various environments tested. The OCP measurements provided initial insights into the electrochemical stability of the metals, Figs 9 and 10 depict the OCP of Q235 carbon steel and galvanized steel. while the polarization curves helped to assess their corrosion rates and the passivation behavior in different conductive concrete mixtures, the polarization curves for Q235 carbon steel and galvanized steel can be seen in Figs 11 and 12. The EIS data further elucidated the charge transfer resistance and capacitive

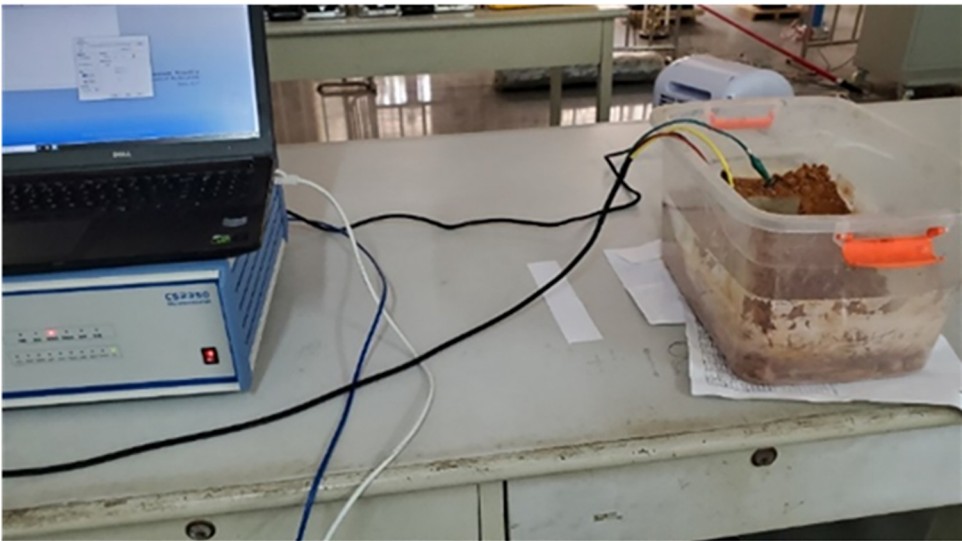

**Fig 8. Connecting scheme of the test platform.**

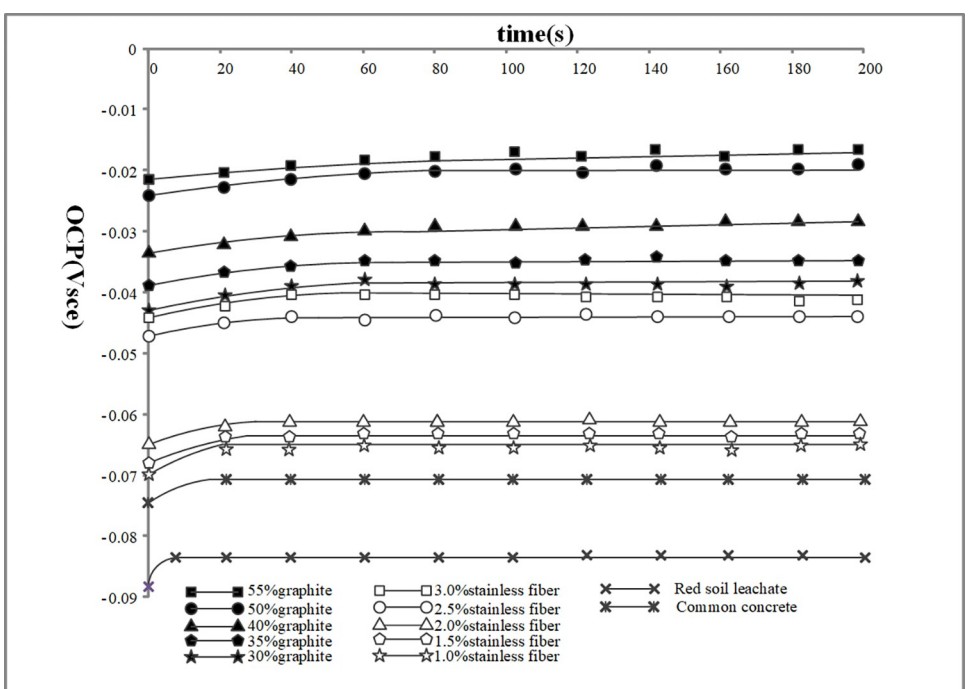

**Fig 9. Open circuit potential test results of Q235 carbon steel.**

behavior of the metals, which are critical parameters for understanding their long-term corrosion resistance, the EIS results are presented in Figs 13 and 14. This comprehensive analysis allowed for the evaluation of how the varying conductive media, specifically the type and amount of conductive filler (graphite or stainless steel fiber), influence the corrosion

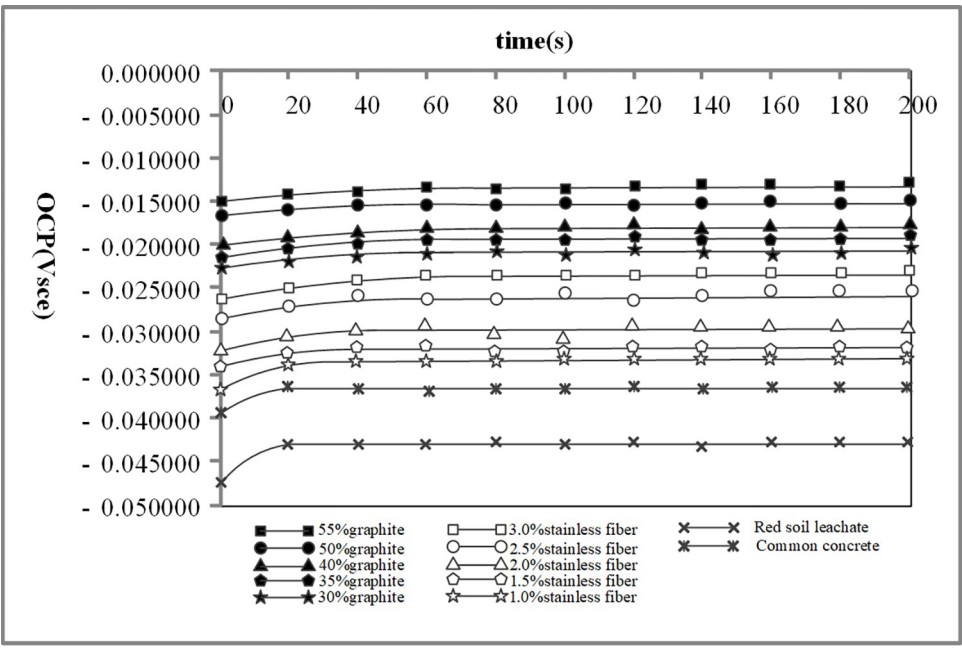

**Fig 10. Open circuit potential test results of galvanized steel.**

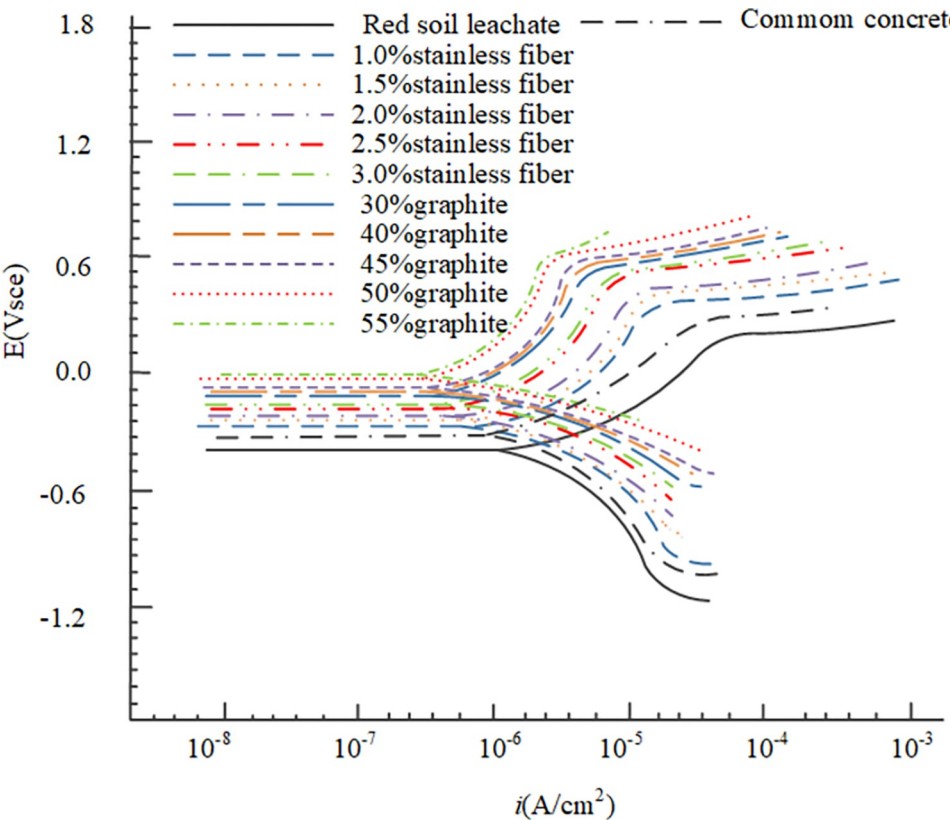

**Fig 11. Polarization curve of Q235 carbon steel.**

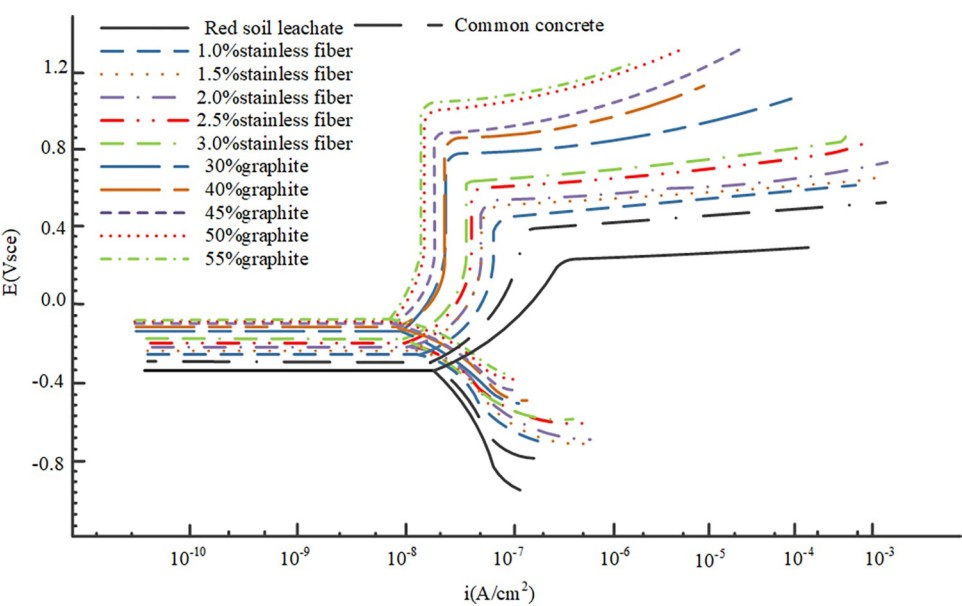

**Fig 12. Polarization curve of galvanized steel.**

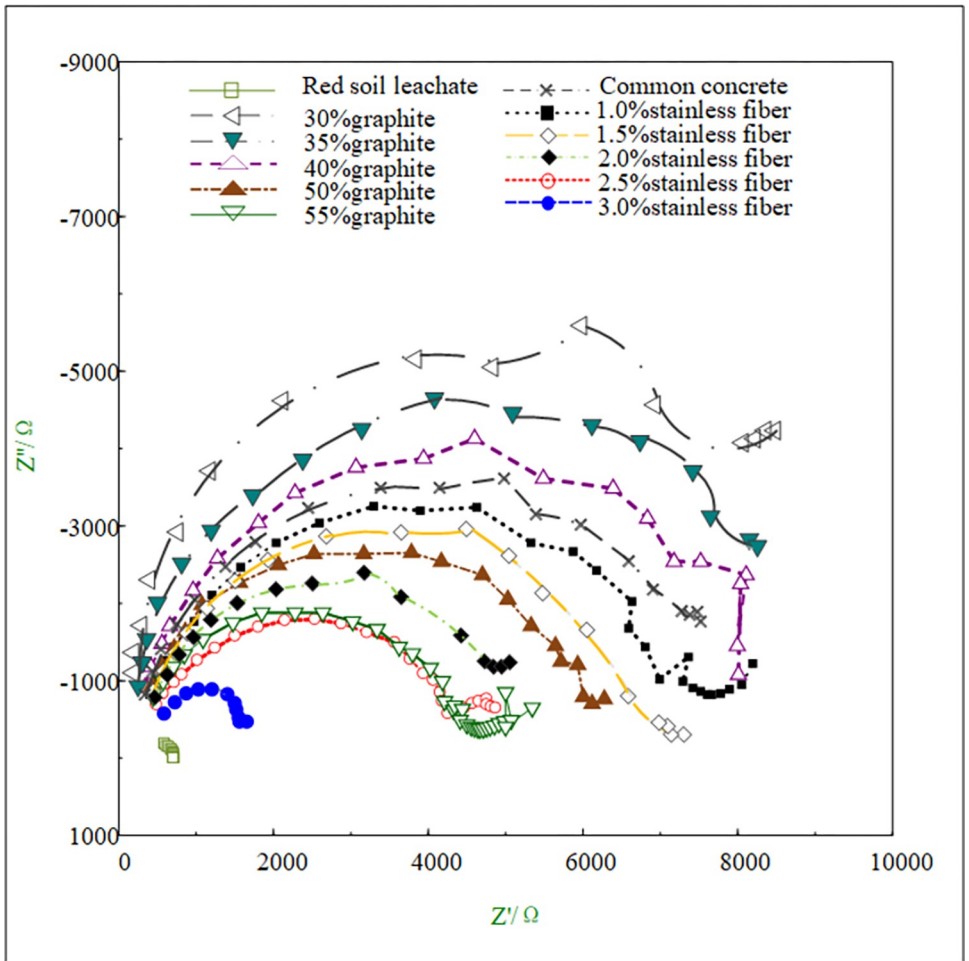

**Fig 13. Electrochemical impedance spectroscopy of Q235 carbon steel.**

performance of the two metals. These findings contribute valuable data for optimizing material selection and design in applications requiring effective grounding and corrosion resistance in diverse environmental conditions. In these Figs, "SSF" refers to Stainless Steel Fiber.

## 4.3. Comparison of metal corrosion in different soil environments

From Figs 9 and 10, it can be clearly seen that the open-circuit potential of metal electrodes in red soil is lower than that in concrete environments, indicating a more aggressive corrosive nature. Conversely, specimens that incorporate conductive phases, such as those with graphite or stainless steel fibers, exhibit higher open-circuit potentials compared to those in ordinary concrete. Specifically, the metal samples that are wrapped in graphite-based or stainless steel fiber-based conductive concrete show a significant increase in their open-circuit potential. These findings demonstrate that the presence of conductive materials in the concrete effectively lowers the corrosion sensitivity of metals, providing an enhanced protective effect against corrosion.

The polarization curves depicted in Figs 11 and 12 provide additional insights into the corrosion behavior in different environments. In red soil, both the corrosion potential is at its lowest and the corrosion current at its highest, indicating the most severe corrosion

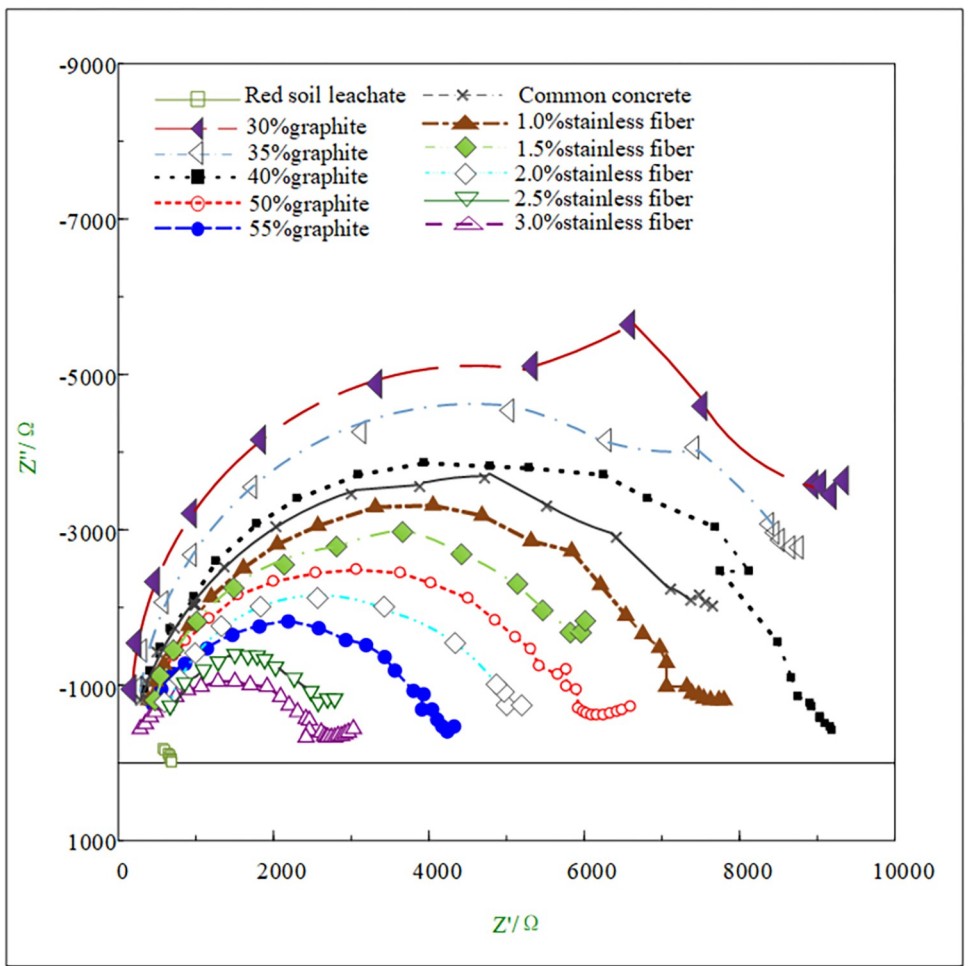

**Fig 14. Electrochemical impedance spectroscopy of galvanized steel.**

conditions. In ordinary concrete, there is an increase in corrosion potential and a corresponding decrease in corrosion current, signifying a reduction in corrosion rate compared to the red soil environment. In the case of conductive concrete, a notable increase in corrosion potential is observed along with a more pronounced decrease in corrosion current, reflecting a further reduction in corrosion rate. Furthermore, the passivation zone—a region where the metal forms a protective oxide layer—is significantly shorter in red soil than in either ordinary or conductive concrete. This suggests that the abundance of acidic ions in red soil accelerates the oxidation of the metal surface, thereby preventing the formation of a stable passivation film. Consequently, wrapping the metal electrode with conductive concrete not only delays the onset of corrosion but also significantly enhances the overall durability of the metal.

The electrochemical impedance spectroscopy (EIS) results shown in Figs 13 and 14 further substantiate these observations. The capacitance arc radius, which indicates the resistance to charge transfer and thus the corrosion resistance, is much smaller in red soil compared to concrete environments. This demonstrates the poor corrosion resistance of grounding metals when exposed to red soil. In contrast, the capacitance arc radius in conductive concrete is generally larger and varies from that in ordinary concrete, depending on the specific type of conductive material used (graphite or stainless steel fibers) and its concentration. This variability highlights the tailored effect of conductive materials on improving corrosion resistance.

In summary, across the four different environmental conditions tested, red soil presents the highest propensity for corrosion, followed by ordinary concrete, whereas conductive concrete exhibits superior corrosion resistance. The primary reason for this difference lies in the significantly higher water content and ion concentration in red soil, which promotes corrosion, whereas ordinary concrete, with its lower moisture and free ion content, demonstrates less corrosive potential. Conductive concrete, enriched with conductive fillers like graphite or stainless steel fibers, not only enhances the dispersion of electrical current during testing but also minimizes electrolytic corrosion under electrification, thereby significantly improving the overall corrosion resistance of the embedded metal. These findings provide valuable insights for optimizing the use of conductive concrete in environments requiring effective grounding and corrosion prevention.

## 4.4. Comparison of metal corrosion in different conductive concrete

Based on the preceding analysis, conductive concrete has been shown to provide effective corrosion protection for embedded metals. This material can incorporate different types of conductive phase materials, such as graphite, which represents non-metallic conductive phases, and stainless steel fiber, which represents metallic conductive phases. To further understand their effects on corrosion protection, a comparative evaluation of these different conductive phases is necessary.

As illustrated in Figs 10 and 11, the open circuit potential of metal electrodes in graphite-based conductive concrete is consistently higher than that in stainless steel fiber-based conductive concrete. This higher potential suggests a more noble behavior and indicates a reduced tendency for corrosion. Furthermore, the self-corrosion potential observed in Figs 11 and 12 is also higher for graphite-based conductive concrete compared to stainless steel fiber-based conductive concrete. This finding reinforces the conclusion that graphite conductive concrete exhibits a lower propensity for corrosion and offers better overall corrosion resistance than stainless steel fiber conductive concrete.

Additionally, electrochemical impedance spectroscopy data, as shown in Figs 13 and 14, further supports these observations. The radius of the capacitance arc, which represents the resistance to charge transfer at the metal-concrete interface, is larger for graphite conductive concrete than for stainless steel fiber conductive concrete. A larger capacitance arc radius signifies greater impedance to corrosive charge movement, thus indicating superior corrosion protection. This result confirms that graphite conductive concrete is less susceptible to corrosion compared to its stainless steel fiber counterpart.

Taken together, all three indicators—open circuit potential, self-corrosion potential, and capacitance arc radius—consistently demonstrate that graphite-based conductive concrete provides enhanced corrosion resistance relative to stainless steel fiber-based conductive concrete. This is likely due to the non-metallic nature of graphite, which contributes to its stable electrochemical behavior, reduced reactivity with the environment, and effective blocking of corrosive ion penetration. In contrast, stainless steel fibers, while conductive, may introduce localized galvanic effects that slightly reduce their overall corrosion resistance. Consequently, graphite-based conductive concrete emerges as a superior choice for applications where enhanced durability and long-term corrosion protection are critical.

## 4.5. Comparison of metal corrosion in different conductive material content

Beyond the type of conductive phase materials used, the content or concentration of these materials also plays a critical role in determining the corrosion resistance of conductive

concrete. As depicted in Figs 9–14, for any given type of conductive phase material—whether graphite or stainless steel fibers—an increase in content leads to a noticeable rise in the open circuit potential, polarization potential, and the capacitance arc radius of the embedded metals. These trends indicate that the corrosion resistance of conductive concrete is enhanced as the concentration of conductive phase materials increases.

From a corrosion resistance standpoint, higher concentrations of conductive materials contribute to improved performance by reducing the rate of electrochemical reactions and increasing the impedance to corrosive charge movement. This is because more conductive materials, whether graphite or stainless steel fibers, help to distribute electrical currents more effectively across the concrete matrix, thereby reducing localized anodic and cathodic sites that can promote corrosion. Additionally, increased conductive content enhances the formation of passivation films on the metal surface, further protecting it from corrosive elements.

However, while higher concentrations of conductive phase materials enhance corrosion protection, practical applications require a balanced approach. Excessive amounts of conductive materials can negatively impact the mechanical strength and structural integrity of the concrete, potentially leading to reduced durability under mechanical loads. Moreover, higher content levels can also affect the overall conductivity and workability of the concrete, making it more challenging to apply or use in construction settings.

Therefore, optimizing the ratio of conductive materials within the concrete mix is crucial to achieving a balance between corrosion resistance, mechanical strength, and overall performance. This requires careful consideration of the specific environmental conditions, desired conductivity levels, and structural requirements. An optimal balance ensures that the conductive concrete provides effective long-term corrosion protection while maintaining adequate mechanical properties and ease of application.

## 4.6. Corrosion comparison of different grounding metals

A comparative analysis of Figs 9 and 10, under the same red soil environment, reveals a significant increase in the open circuit potential of galvanized steel compared to Q235 carbon steel. For galvanized steel, the potential stabilizes at a more positive value, with the lowest recorded value shifting from -0.083V for Q235 carbon steel to -0.0475V. This positive shift indicates that galvanized steel has a lower corrosion sensitivity and enhanced corrosion resistance in the red soil environment.

Further insights can be drawn from the polarization curves shown in Figs 11 and 12, where the self-corrosion current of galvanized steel is notably lower than that of Q235 carbon steel, with a reduction of approximately two orders of magnitude. This suggests a significant decrease in the rate of metal dissolution for galvanized steel. Additionally, the slope of the passivation zone for galvanized steel is steeper and more defined, indicating that the protective zinc coating effectively promotes the formation of a stable passivation layer, which provides superior corrosion resistance when compared to the bare surface of Q235 carbon steel.

Figs 13 and 14 illustrate the electrochemical impedance spectra for both metals in red soil solution, showing similar general patterns. However, when conductive phase materials are added, the capacitance arc radius for galvanized steel is observed to be slightly larger than that of Q235 carbon steel. A larger capacitance arc radius suggests higher charge transfer resistance, further confirming the superior corrosion resistance of galvanized steel.

Collectively, these observations highlight that galvanized steel exhibits better corrosion resistance than Q235 carbon steel across various environmental conditions. The protective zinc layer on galvanized steel acts as a sacrificial anode, mitigating corrosion by providing a first line of defense against corrosive elements and reducing the electrochemical activity on the

steel surface. This makes galvanized steel a more robust choice for grounding applications, especially in environments where high corrosion resistance is critical.

## 5. Conclusion

1. The study reveals that red soil significantly accelerates the corrosion of grounding metals buried within it, while conductive concrete proves to be an effective material for corrosion protection. Measurements from open circuit potential, dynamic polarization curves, and electrochemical impedance spectroscopy consistently indicate that corrosion is slowest in conductive concrete, with a moderate rate observed in ordinary concrete, and the highest rate occurring in red soil. These findings highlight the importance of selecting suitable materials to mitigate corrosion in aggressive soil environments.

2. Among the different types of conductive concrete tested, those containing graphite as the conductive phase material demonstrate superior corrosion protection compared to those incorporating stainless steel fibers. This is evidenced by the higher open circuit potentials, lower corrosion currents, and larger capacitance arc radii associated with graphite-based conductive concrete, all of which point to its enhanced ability to resist corrosion. The non-metallic nature of graphite appears to contribute to its stability and effectiveness in minimizing corrosive interactions at the metal-concrete interface.

3. The results further show that increasing the content of conductive phase materials, such as graphite or stainless steel fibers, in conductive concrete improves its corrosion protection capabilities. Higher concentrations of these materials lead to greater open circuit potentials, increased polarization potentials, and larger capacitance arc radii, all of which are indicative of enhanced corrosion resistance. However, practical applications must strike a balance between the concentration of conductive materials and the mechanical strength, workability, and overall conductivity of the concrete to ensure optimal performance.

4. Galvanized steel exhibits significantly stronger corrosion resistance and lower corrosion rates compared to Q235 carbon steel when encased in conductive concrete. This is particularly evident in environments such as red soil, where galvanized steel shows a more positive open circuit potential, lower self-corrosion currents, and more robust passivation behavior. The zinc coating on galvanized steel provides a sacrificial layer that effectively mitigates corrosion, making it a superior choice for grounding applications. Therefore, it is recommended to use galvanized steel encased in conductive concrete for grounding in red soil areas within power systems to maximize durability and minimize maintenance costs.

## Supporting information

**S1 File.**
(XLSX)

## Author Contributions

**Conceptualization:** Shan Lin, Feng Pei, Hongbo Cheng.

**Data curation:** Pinghao Yang.

**Funding acquisition:** Hongbo Cheng.

**Investigation:** Feng Pei.

**Methodology:** Shan Lin, Xu Tian, Feng Pei.

**Project administration:** Hongbo Cheng.

**Resources:** Shan Lin, Xu Tian, Feng Pei.

**Software:** Pinghao Yang.

**Validation:** Shan Lin, Feng Pei.

**Visualization:** Xu Tian.

**Writing – original draft:** Pinghao Yang.

**Writing – review & editing:** Hongbo Cheng.

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
