## [Decision Letter · Decision Letter 0]

23 Jul 2024

PONE-D-24-25823Comparative Study on Corrosion Characteristics of Conductive Concrete in Red Soil EnvironmentPLOS ONE

Dear Dr. Cheng,

Thank you for submitting your manuscript to PLOS ONE. After careful consideration, we feel that it has merit but does not fully meet PLOS ONE’s publication criteria as it currently stands. Therefore, we invite you to submit a revised version of the manuscript that addresses the points raised during the review process.

We look forward to receiving your revised manuscript.

Kind regards,

Din Bandhu, Ph.D.

Academic Editor

PLOS ONE

Journal Requirements:

   "This research was funded by the Key projects of Natural Science Foundation of Jiangxi Province under grant 2021ACB204004."

4. In this instance it seems there may be acceptable restrictions in place that prevent the public sharing of your minimal data. However, in line with our goal of ensuring long-term data availability to all interested researchers, PLOS’ Data Policy states that authors cannot be the sole named individuals responsible for ensuring data access (http://journals.plos.org/plosone/s/data-availability#loc-acceptable-data-sharing-methods).

Reviewers' comments:

Reviewer's Responses to Questions

**Comments to the Author**

1. Is the manuscript technically sound, and do the data support the conclusions?

Reviewer #1: Yes

Reviewer #2: Yes

2. Has the statistical analysis been performed appropriately and rigorously? 

Reviewer #1: Yes

Reviewer #2: Yes

3. Have the authors made all data underlying the findings in their manuscript fully available?

Reviewer #1: Yes

Reviewer #2: Yes

4. Is the manuscript presented in an intelligible fashion and written in standard English?

Reviewer #1: Yes

Reviewer #2: Yes

5. Review Comments to the Author

Reviewer #1: --- The following list of comments will help to further improve the manuscript:

- The most striking section of a study is the abstract section. Therefore, important results from the study should be highlighted.

- The novelty of the study should be mentioned a little more in the introduction section.

- The flowchart should be added to the study. Thus, the intelligibility of the study will increase.

- The “Grounding Metal Corrosion Evaluation Index and Test Method” and “Testing and Analysis of Corrosion Characteristics of Conductive Concrete Grounding Materials” sections should be rewritten more comprehensively.

- The following sections should be rewritten carefully.

“4.2 Grounding Metals Corrosion Characteristics Testing in Different Environments The corrosion characteristics of commonly used grounding metals, Q235 carbon steel and galvanized steel, in red soil, ordinary concrete, and conductive concrete were investigated using the CS310H electrochemical workstation. Open-circuit potential, potentiodynamic polarization curves, and electrochemical impedance spectroscopy were conducted and analyzed for these metals in various environments. Different types and compositions of conductive concrete were also tested to evaluate their impact on the corrosion behavior of grounding metals. Fig 8 and Fig 9 depict the open-circuit potentials of Q235 carbon steel (comprising C, Mn, Si, S, P, and Fe) and galvanized steel (composed of common carbon steel with a zinc and aluminum coating). The polarization curves for Q235 carbon steel and galvanized steel can be seen in Fig 10 and Fig 11, while electrochemical impedance spectroscopy results are presented in Fig 12 and Fig 13. In these Figs, "SSF" refers to Stainless Steel Fiber.

4.3 Comparison of Metal Corrosion in Different Soil Environments From Fig 8 and Fig 9, it can be seen that the open-circuit potential of metal electrodes in red soil is lower than that in concrete, and the potential of specimens with conductive phase is higher than that of the ordinary concrete. The open circuit potential of metal samples was significantly increased whether they were wrapped in graphite based or stainless steel fiberbased conductive concrete. The results show that the corrosion sensitivity of metals in conductive concrete is greatly reduced. From the polarization curves in Fig 10 and Fig 11, it is evident that in red soil, the corrosion potential is the lowest and the corrosion current is the highest. In ordinary concrete, the corrosion potential increases and the corrosion current decreases compared to red soil. In conductive concrete, there is a noticeable increase in corrosion potential and a more significant decrease in corrosion current. Additionally, the passivation zone in red soil is notably shorter than in ordinary concrete and conductive concrete. This suggests that the abundance of acidic ions in red soil accelerates surface oxidation of the metal, making it challenging to form a stable passivation film. Therefore, wrapping the metal electrode with conductive concrete can potentially delay metal corrosion and enhance its durability. From the electrochemical impedance spectroscopy shown in Fig 12 and Fig 13, it is apparent that compared to concrete, the capacitance arc radius in red soil is very small, indicating poor corrosion resistance of grounding metals in a red soil environment. Furthermore, the capacitance arc radius in conductive concrete differs from that in ordinary concrete, depending on the specific conductive materials added and their concentration. In summary, among the four different soil environments tested, red soil exhibits the highest corrosion propensity, followed by ordinary concrete, while conductive concrete demonstrates good corrosion resistance. This difference arises due to the higher water and ion concentrations in red soil and the lower moisture and free ion concentrations in concrete, thereby weakening concrete's corrosion capability. Conductive concrete, enriched with conductive materials, serves to shunt current during experiments, thereby reducing electrolytic corrosion during electrification processes. 4.4 Comparison of Metal Corrosion in Different Conductive Concrete Based on the previous analysis, conductive concrete demonstrates effective corrosion protection. This material incorporates various conductive phase materials, with graphite representing non-metallic conductive phases and stainless steel fiber representing metallic conductive phases. Let's compare the effects of these different conductive phase materials on corrosion protection. In Figs 9 and 10, the open circuit potential of graphite conductive concrete is higher compared to stainless steel fiber conductive concrete. Additionally, the self-corrosion potential of graphite conductive concrete in Figs 10 and 11 is also higher than that of stainless steel fiber conductive concrete, indicating that graphite conductive concrete exhibits a smaller corrosion tendency and better corrosion resistance than stainless steel fiber. Moreover, in Figs 12 and 13, the radius of the capacitance arc for graphite conductive concrete is larger than that for stainless steel conductive concrete. This indicates that graphite conductive concrete offers greater impedance to corrosive charges and is less prone to corrosion. All three indicators consistently suggest that graphite conductive concrete provides superior corrosion resistance compared to stainless steel fiber conductive concrete. 4.5 Comparison of Metal Corrosion in Different Conductive Material Content In addition to the type of conductive phase materials, the content of these materials also significantly impacts the corrosion resistance of conductive concrete. Figs 8 to 13 illustrate that, for a given conductive phase material, increasing its content results in higher open circuit potential, polarization potential, and capacitance arc radius of the metal. These observations indicate that the corrosion resistance of conductive concrete improves as the content of conductive phase materials increases From a corrosion resistance perspective, higher concentrations of conductive phase materials positively affect the performance of conductive concrete. However, in practical applications, it is crucial to balance the content of conductive phase materials with considerations of mechanical strength and conductivity. Achieving an optimal ratio is essential to ensure both effective corrosion protection and satisfactory overall performance of conductive concrete. 4.6 Corrosion Comparison of Different Grounding Metals Comparing Fig 8 and Fig 9 under the same red soil environment, it is evident that the open circuit potential of galvanized steel has significantly increased compared to Q235 carbon steel. Upon stabilization, the potential becomes more positive, with the lowest value rising from - 0.083V to -0.0475V. This shift indicates lower corrosion sensitivity and stronger corrosion resistance for galvanized steel. In Figs 10 and 11, the self-corrosion current of galvanized steel is slightly positive compared to Q235 carbon steel, with a decrease of approximately two orders of magnitude in corrosion current. The passivation zone slope is also steeper and more pronounced for galvanized steel, suggesting that the zinc coating provides better corrosion resistance compared to Q235 carbon steel. Figs 12 and 13 show that the electrochemical impedance spectra of galvanized steel in red soil solution are generally similar to those of Q235 carbon steel. However, upon adding conductive phase materials, the capacitance arc radius of galvanized steel is slightly larger than that of Q235 carbon steel. These observations collectively indicate that galvanized steel exhibits better corrosion resistance than Q235 carbon steel across various environmental conditions.”

- Lines in graphs should be thickened. (too bad)

- The “Conclusion” section should be reviewed. (It should be enriched with the results obtained.)

Reviewer #2: Overall, this is a well-written manuscript. The introduction is relevant and theory-based, supported by experiments. Sufficient information about the previous study findings is presented for readers. The methods are generally appropriate Overall, the results are clear because of the good graphical representation.

6. PLOS authors have the option to publish the peer review history of their article (what does this mean?). If published, this will include your full peer review and any attached files.

Reviewer #1: No

Reviewer #2: No

---

## [Author Response · Author response to Decision Letter 0]

9 Sep 2024

Reviewer #1: --- The following list of comments will help to further improve the manuscript:

- The most striking section of a study is the abstract section. Therefore, important results from the study should be highlighted.

Response: we have modified the abstract.

- The novelty of the study should be mentioned a little more in the introduction section.

Response: we have added a paragraph in introduction to address our novelty, which is highlighted.

- The flowchart should be added to the study. Thus, the intelligibility of the study will increase.

Response: that’s a good idea, and we have draw a diagram to show the diagram of our experiment, shown as Fig 5.

- The “Grounding Metal Corrosion Evaluation Index and Test Method” and “Testing and Analysis of Corrosion Characteristics of Conductive Concrete Grounding Materials” sections should be rewritten more comprehensively.

Response: we have added and revised in section 4.

- The following sections should be rewritten carefully.

“4.2 Grounding Metals Corrosion Characteristics Testing in Different Environments The corrosion characteristics of commonly used grounding metals, Q235 carbon steel and galvanized steel, in red soil, ordinary concrete, and conductive concrete were investigated using the CS310H electrochemical workstation. Open-circuit potential, potentiodynamic polarization curves, and electrochemical impedance spectroscopy were conducted and analyzed for these metals in various environments. Different types and compositions of conductive concrete were also tested to evaluate their impact on the corrosion behavior of grounding metals. Fig 8 and Fig 9 depict the open-circuit potentials of Q235 carbon steel (comprising C, Mn, Si, S, P, and Fe) and galvanized steel (composed of common carbon steel with a zinc and aluminum coating). The polarization curves for Q235 carbon steel and galvanized steel can be seen in Fig 10 and Fig 11, while electrochemical impedance spectroscopy results are presented in Fig 12 and Fig 13. In these Figs, "SSF" refers to Stainless Steel Fiber.

4.3 Comparison of Metal Corrosion in Different Soil Environments From Fig 8 and Fig 9, it can be seen that the open-circuit potential of metal electrodes in red soil is lower than that in concrete, and the potential of specimens with conductive phase is higher than that of the ordinary concrete. The open circuit potential of metal samples was significantly increased whether they were wrapped in graphite based or stainless steel fiberbased conductive concrete. The results show that the corrosion sensitivity of metals in conductive concrete is greatly reduced. From the polarization curves in Fig 10 and Fig 11, it is evident that in red soil, the corrosion potential is the lowest and the corrosion current is the highest. In ordinary concrete, the corrosion potential increases and the corrosion current decreases compared to red soil. In conductive concrete, there is a noticeable increase in corrosion potential and a more significant decrease in corrosion current. Additionally, the passivation zone in red soil is notably shorter than in ordinary concrete and conductive concrete. This suggests that the abundance of acidic ions in red soil accelerates surface oxidation of the metal, making it challenging to form a stable passivation film. Therefore, wrapping the metal electrode with conductive concrete can potentially delay metal corrosion and enhance its durability. From the electrochemical impedance spectroscopy shown in Fig 12 and Fig 13, it is apparent that compared to concrete, the capacitance arc radius in red soil is very small, indicating poor corrosion resistance of grounding metals in a red soil environment. Furthermore, the capacitance arc radius in conductive concrete differs from that in ordinary concrete, depending on the specific conductive materials added and their concentration. In summary, among the four different soil environments tested, red soil exhibits the highest corrosion propensity, followed by ordinary concrete, while conductive concrete demonstrates good corrosion resistance. This difference arises due to the higher water and ion concentrations in red soil and the lower moisture and free ion concentrations in concrete, thereby weakening concrete's corrosion capability. Conductive concrete, enriched with conductive materials, serves to shunt current during experiments, thereby reducing electrolytic corrosion during electrification processes. 4.4 Comparison of Metal Corrosion in Different Conductive Concrete Based on the previous analysis, conductive concrete demonstrates effective corrosion protection. This material incorporates various conductive phase materials, with graphite representing non-metallic conductive phases and stainless steel fiber representing metallic conductive phases. Let's compare the effects of these different conductive phase materials on corrosion protection. In Figs 9 and 10, the open circuit potential of graphite conductive concrete is higher compared to stainless steel fiber conductive concrete. Additionally, the self-corrosion potential of graphite conductive concrete in Figs 10 and 11 is also higher than that of stainless steel fiber conductive concrete, indicating that graphite conductive concrete exhibits a smaller corrosion tendency and better corrosion resistance than stainless steel fiber. Moreover, in Figs 12 and 13, the radius of the capacitance arc for graphite conductive concrete is larger than that for stainless steel conductive concrete. This indicates that graphite conductive concrete offers greater impedance to corrosive charges and is less prone to corrosion. All three indicators consistently suggest that graphite conductive concrete provides superior corrosion resistance compared to stainless steel fiber conductive concrete. 4.5 Comparison of Metal Corrosion in Different Conductive Material Content In addition to the type of conductive phase materials, the content of these materials also significantly impacts the corrosion resistance of conductive concrete. Figs 8 to 13 illustrate that, for a given conductive phase material, increasing its content results in higher open circuit potential, polarization potential, and capacitance arc radius of the metal. These observations indicate that the corrosion resistance of conductive concrete improves as the content of conductive phase materials increases From a corrosion resistance perspective, higher concentrations of conductive phase materials positively affect the performance of conductive concrete. However, in practical applications, it is crucial to balance the content of conductive phase materials with considerations of mechanical strength and conductivity. Achieving an optimal ratio is essential to ensure both effective corrosion protection and satisfactory overall performance of conductive concrete. 4.6 Corrosion Comparison of Different Grounding Metals Comparing Fig 8 and Fig 9 under the same red soil environment, it is evident that the open circuit potential of galvanized steel has significantly increased compared to Q235 carbon steel. Upon stabilization, the potential becomes more positive, with the lowest value rising from - 0.083V to -0.0475V. This shift indicates lower corrosion sensitivity and stronger corrosion resistance for galvanized steel. In Figs 10 and 11, the self-corrosion current of galvanized steel is slightly positive compared to Q235 carbon steel, with a decrease of approximately two orders of magnitude in corrosion current. The passivation zone slope is also steeper and more pronounced for galvanized steel, suggesting that the zinc coating provides better corrosion resistance compared to Q235 carbon steel. Figs 12 and 13 show that the electrochemical impedance spectra of galvanized steel in red soil solution are generally similar to those of Q235 carbon steel. However, upon adding conductive phase materials, the capacitance arc radius of galvanized steel is slightly larger than that of Q235 carbon steel. These observations collectively indicate that galvanized steel exhibits better corrosion resistance than Q235 carbon steel across various environmental conditions.”

Response: we have rewrited these sections as the content highlighted.

- Lines in graphs should be thickened. (too bad)

Response: we have thickened the lines from Fig 9 to Fig 14.

- The “Conclusion” section should be reviewed. (It should be enriched with the results obtained.)

Response: we have rewrited the conclusion.

Reviewer #2: Overall, this is a well-written manuscript. The introduction is relevant and theory-based, supported by experiments. Sufficient information about the previous study findings is presented for readers. The methods are generally appropriate Overall, the results are clear because of the good graphical representation.

Response: Thank you for your positive feedback and encouraging comments. Your recognition of our work is greatly appreciated.

---

## [Decision Letter · Decision Letter 1]

22 Oct 2024

Comparative Study on Corrosion Characteristics of Conductive Concrete in Red Soil Environment

PONE-D-24-25823R1

Dear Dr. Cheng,

We’re pleased to inform you that your manuscript has been judged scientifically suitable for publication and will be formally accepted for publication once it meets all outstanding technical requirements.

Kind regards,

Din Bandhu, Ph.D.

Academic Editor

PLOS ONE

Additional Editor Comments (optional):

Reviewers' comments:

Reviewer's Responses to Questions

**Comments to the Author**

1. If the authors have adequately addressed your comments raised in a previous round of review and you feel that this manuscript is now acceptable for publication, you may indicate that here to bypass the “Comments to the Author” section, enter your conflict of interest statement in the “Confidential to Editor” section, and submit your "Accept" recommendation.

Reviewer #1: All comments have been addressed

2. Is the manuscript technically sound, and do the data support the conclusions?

Reviewer #1: Yes

3. Has the statistical analysis been performed appropriately and rigorously? 

Reviewer #1: Yes

4. Have the authors made all data underlying the findings in their manuscript fully available?

Reviewer #1: Yes

5. Is the manuscript presented in an intelligible fashion and written in standard English?

Reviewer #1: Yes

6. Review Comments to the Author

Reviewer #1: Manuscript ID: PONE-D-24-25823R1 entitled " Comparative Study on Corrosion Characteristics of Conductive Concrete in Red Soil Environment" for journal of “PLOS ONE" has been reviewed.

After the revision carried out by the authors;

-Abstract clearly presents objects methods and results.

-Keywords are adequate.

-Scientific methods are adequately used.

-Terminology is adequate.

-Results are clearly presented.

-Conclusions are logically derived from the data presented.

----

Consequently;

The authors have made significant improvements to the paper by addressing the feedback provided by the reviewers, resulting in a clearer presentation of results. Based on these revisions, the paper is now ready for acceptance.

7. PLOS authors have the option to publish the peer review history of their article (what does this mean?). If published, this will include your full peer review and any attached files.

Reviewer #1: No

---

## [Editor Report · Acceptance letter]

28 Oct 2024

PONE-D-24-25823R1 

PLOS ONE

Dear Dr. Cheng, 

I'm pleased to inform you that your manuscript has been deemed suitable for publication in PLOS ONE. Congratulations! Your manuscript is now being handed over to our production team.

Kind regards, 

on behalf of

Dr. Din Bandhu 

Academic Editor

PLOS ONE